# The Role of Neutrophils in Hypertension

**DOI:** 10.3390/ijms21228536

**Published:** 2020-11-12

**Authors:** Patricio Araos, Stefanny Figueroa, Cristián A. Amador

**Affiliations:** Laboratory of Renal Physiopathology, Institute of Biomedical Sciences, Universidad Autónoma de Chile, Santiago 8910060, Chile; patricio.araos@gmail.com (P.A.); smfigueroar@gmail.com (S.F.)

**Keywords:** neutrophil, hypertension, oxidative stress, inflammation, innate immunity

## Abstract

It is well accepted that the immune system and some cells from adaptive and innate immunity are necessary for the initiation/perpetuation of arterial hypertension (AH). However, whether neutrophils are part of this group remains debatable. There is evidence showing that the neutrophil/lymphocyte ratio correlates with AH and is higher in non-dipper patients. On the other hand, the experimental neutrophil depletion in mice reduces basal blood pressure. Nevertheless, their participation in AH is still controversial. Apparently, neutrophils may modulate the microenvironment in blood vessels by increasing oxidative stress, favoring endothelial disfunction. In addition, neutrophils may contribute to the tissue infiltration of immune cells, secreting chemoattractant chemokines/cytokines and promoting the proinflammatory phenotype, leading to AH development. In this work, we discuss the potential role of neutrophils in AH by analyzing different mechanisms proposed from clinical and basic studies, with a perspective on cardiovascular and renal damages relating to the hypertensive phenotype.

## 1. Introduction

Arterial hypertension (AH) is a worldwide health problem and a huge economic burden in developing and developed nations, where the projected cost for 2030 in the United States of America (USA), for example, could reach values upward of USD 200 billion [1]. AH affects 31.1% of the population [2] and its inappropriate handling represents the leading risk factor for mortality worldwide [3,4]. In this sense, AH can promote chronic kidney disease, another critical cardiovascular (CV) risk factor, affecting ~13% of the population, and this remains one of the leading causes for progression to end-stage renal disease requiring renal replacement therapies. Therefore, AH, renal, and CV diseases are complex conditions with a demonstrated causal relationship, supporting the need of new alternatives for earlier detection and adequate treatment.

Above 70% of AH patients have no etiology and they remain categorized as “essential” or “primary”. The origin of this situation is complex and multifactorial, and it involves the interaction of several physiological systems boosted by exposure to lifestyle risk factors [2]. Recently, new mechanisms associated with the onset of primary AH and its progression to renal and CV diseases have been discussed, where the immune system and its activation may be critical.

The available evidence strongly indicates that immune cells can modulate cardiac output and vascular resistance causing AH, fibrosis, inflammation, oxidative stress, and organ damage [5]. Here, we discuss the effect of immune system activation, particularly the role of neutrophils in the generation and maintenance of AH through the promotion of different mechanisms that permit hypothesizing their direct and indirect contribution to renal and CV damage.

## 2. Role of Immune System in Hypertension

The evidence linking the immune system and AH dates from the 1960s [6,7]; however, the hypothesis of a causal role of the immune system in AH was proposed years later by Okuda and Grollman, where the transference of lymph node cells from rats with unilateral renal artery ligation induced AH in recipient rats [8]. Likewise, splenocytes from rats treated with deoxycorticosterone acetate (DOCA) salt induced AH in normotensive animals [9]. Conversely, the absence of thymus was sufficient to protect against AH development in the DOCA salt and genetic models [10,11], while thymus grafting from wild-type mice (WT) into nude mice (resistant to AH generation) restored their sensitivity to DOCA salt-dependent AH [10]. In this context, the most recent evidence suggests that the autonomic nervous system may control the immune system [12], since there is a neuroimmune boost at the splenic level during AH [13]. This emerging evidence proposes the autonomic nervous system as a modulator of the immune system in AH, beyond its classic effects on heart rate, vascular tone, and sodium excretion.

Guzik et al. found that T lymphocytes and not B lymphocytes were necessary for AH development and vascular dysfunction, as a result of DOCA salt or angiotensin II (AngII) [14]. Years later, these findings were confirmed by other groups and in other experimental settings of AH that included salt sensitivity [15,16]. In addition, it was suggested that T lymphocytes can be activated by a cholinergic impulse, through a vagus nerve connection to the spleen, promoting their migration to the target organs and contributing to AH [13].

In the last 10 years, different lymphocytes have been studied in order to establish their role in blood pressure control. On the adaptive immunity side, CD8^+^ T cells apparently participate in AH by increasing renal sodium reabsorption through the activation of specific transporters [17,18]. On the other hand, CD4^+^ lymphocytes, both T helper 1 (T_H_1) cells and T_H_17 cells, accumulate in the vessels, spleen, and kidney during AH [19,20]. Causal involvement of CD4^+^ cells was demonstrated by blood pressure modulation in various studies using genetically lacking CD4^+^ lymphocytes animals or knockout (KO) animals for cytokines from T_H_1 or T_H_17 cells [21,22,23]. However, since other cells can also secrete these cytokines, the involvement of T_H_1 and T_H_17 has not been entirely clarified. Contrarily, it has been proposed that the activity of regulatory T (Treg) cells, a subpopulation of CD4^+^ lymphocytes, could be decreased in AH. Different studies have shown a decreasing Treg cell population in plasma, spleen, or kidney during AH, while the selective depletion of Treg cells raises blood pressure and exacerbates target organ damage in experimental models [24,25]. In all these studies, the adoptive transfer (or activation) of Treg cells prevented AH and CV damage in response to AngII or aldosterone, supporting their promissory use in cell therapy strategies.

Regarding the innate immune cells, unconventional γδ T cells that present a rapid innate-like response initiating immune response are involved in AH development by regulating vasoconstriction and by secreting large amounts of interleukin (IL)-17 [26]. These observations delineated new perspectives for IL-17, beyond T_H_17 lymphocytes. On the other hand, dendritic cells (DCs) are considered as critical for the AH since there is evidence showing that they participate in T_H_1 and T_H_17 cell activation, by taking up neoantigens and through isoketal formation after Nicotinamide adenine dinucleotide phosphate (NADPH) oxidase activation [27,28]. Moreover, van Beusecum et al. demonstrated that the specific deletion of serum glucocorticoid kinase 1 (SGK1) in DCs prevents AH, vascular dysfunction, and kidney inflammation induced by *N*-Nitro-l-arginine methyl ester (L-NAME) plus a high-salt diet [29]. Apparently, SGK1 may function as a sodium sensor, allowing DC activation; this effect was also observed in CD4^+^ lymphocytes [30]. The orchestrated responses by DCs are supported by a recent study showing that the absence of DCs protects against renal dysfunction, kidney damage, and CV inflammation/fibrosis in an AngII-dependent hypertensive model [31]. The protective effect of DC depletion for AH development was also corroborated in the nephrectomy aldosterone salt (NAS) model, which involved the prevention of cardiac hypertrophy and fibrosis [32]. Finally, monocytes and macrophages have also been implicated, principally due to their increase and the phenotypic changes in vasculature, kidney, heart, and brain during AH [33], promoting fibrosis and maintaining AH through NADPH oxidase-dependent mechanisms in vascular tissue [34,35]. In general, the accumulative evidence suggests that innate and adaptive immunity are involved in the pathophysiology of AH and in associated end-organ damage. However, the specific immune cells that promote AH and tissue damage are still under study, where the role of neutrophils remains to be addressed.

## 3. Neutrophils as Contributors of High Blood Pressure

Neutrophils are polymorphonuclear granulocytes that represent the most abundant leukocyte in the blood. They are effector cells of the innate immune system with resistance to extracellular pathogens in acute inflammation as their main role. Neutrophils are characterized by their action as phagocytic cells leading to release of the content from their granules, such as proteolytic enzymes and reactive oxygen species (ROS). These properties configure an antimicrobial effect, where the response extension has not been totally clarified due to there being growing evidence showing that neutrophils are able to activate other cell types also involved in the defense against pathogens [36]. Moreover, these characteristics suggest that neutrophils may also act in the sterile immune response. In fact, their participation in CV diseases that involve chronic inflammation, such as atherosclerosis, ischemic stroke, and myocardial infarction has been demonstrated [37].

The discussion concerning the potential contribution of neutrophils in AH seems to have foundation, in part, due to observations in patients showing that elevated blood neutrophil count and neutrophil/lymphocyte ratio (NLR) are closely associated with an increased risk of developing AH [38,39] (Table 1). Similarly, non-dipper hypertensive patients have an increase of ~72% in NLR value compared to dipper patients [40]. In the same way, Belen et al. found an increased NLR in patients with resistant AH compared to patients with controlled AH (systolic blood pressure = 124.3 ± 5.2 mmHg) [41]. In this study, both resistant AH and controlled AH patients presented a significant increase in the NLR when compared to normotensive patients, which may be explained as the effect of differential regulations of antihypertensive treatments on the NLR. For instance, in a double-blind randomized prospective study carried out by Fici et al., it was shown that nebivolol, a selective β1 blocker, reduced blood pressure, prevented vascular microinflammation, and reduced the NLR in an independent way and differently from metoprolol [42]. Similarly, Karaman et al. observed that valsartan, an AngII receptor blocker, was more efficient in reducing the NLR after 12 weeks of treatment compared to amlodipine, a dihydropyridine calcium channel blocker [43].

In a cross-sectional study with 33 diagnosed patients, Aydin et al. showed that patients with a normal–high AH grade belonging to the higher tercile for blood pressure presented a significant increase in neutrophil count and in NLR [49]. On the other hand, Bozduman et al., in a single-center retrospective study with 104 dipper patients and with “normal–high” blood pressure, did not find any increase in NLR compared to normotensive patients [48]. Differences in studied populations and methodological aspects may account for the differences found in both studies. For instance, in the study of Bozduman et al., the cohort of dipper normotensive patients presented a slightly elevated value for systolic blood pressure (116.5 ± 7.5 mmHg), body mass index, and smoking percentage, compared to the patients considered in the study of Aydin et al. (107 ± 5 mmHg), which could in part explain these differences.

Recently, Siedlinski et al. observed an association between AH and the quintiles of counts of white blood cell subpopulations by using mendelian randomization in British patients from the United Kingdom (UK) Biobank, in order to make potentially causal deductions [56]. In particular, they found that the association of blood neutrophil count with systolic, diastolic, and pulse pressure indices was the strongest compared to the other white blood cells analyzed. However, they found that lymphocytes but not neutrophils were causally related to blood pressure levels [56]. Thus, even when this work did not find causality among neutrophils, the authors discussed that their results were susceptible to errors induced by confounding or reverse causation phenomena, which motivates future prospective studies or randomized controlled trials in order to elucidate this question.

It is worth noting that the NLR has been linked to a high probability of mortality in elderly patients with AH [50], and it is also used as a predictive factor for other CV diseases such as acute ischemic stroke [57], epicardial fat tissue thickness [58], and atherosclerosis [59]. However, the pathophysiological significance of neutrophil accumulation (represented as neutrophil count or NLR) and the signaling mechanisms for CV diseases remain unknown and deserve to be explored. At the present time, experimental evidence suggests that neutrophils may participate in AH, principally through the mechanisms that we detail below.

### 3.1. Neutrophils Can Modulate Oxidative Stress and Vascular Response

Oxidative stress is characterized by excessive ROS production, which in turn triggers multiple processes, such as protein oxidation, inflammation, proliferation, and fibrosis, impairing vascular function and promoting CV remodeling.

Under pathological conditions like AH, the imbalance between pro- and antioxidant molecules favor pro-oxidant species, leading to vascular damage [60]. It has been demonstrated that a sustained increase in blood pressure induces an endothelial dysfunction via vascular ROS increment [61], which can also be determined in physiological conditions related to AH, such as aging and pregnancy [62]. Different studies show that neutrophils isolated from the peripheral blood of hypertensive patients, women with preeclampsia, and an experimental AH model present high levels of ROS with a mayor phagocytic activity [44,45,46]. This scenario acquires relevance considering that neutrophils adhere to endothelial cells during host-defense reactions, which include ROS generation via NADPH oxidase and myeloperoxidase (MPO) activation, secretion of proinflammatory mediators, and production of neutrophil extracellular traps formed by DNA fibers and proteins from secretory granules, promoting cellular permeability and vascular dysfunction [63] (Figure 1). Additionally, sympathetic activation can also modulate neutrophil function [64]. Nicholls et al. showed that the incubation of neutrophils with norepinephrine increased, in a dose-dependent manner, the metabolic activity and the MPO and IL-6 release [65]. The possible mechanisms involving the crosstalk of endothelial cells and neutrophils and of norepinephrine and neutrophils remain unknown and may represent future clues for the understanding of blood pressure baseline control.

One experimental study that strongly suggested that neutrophils are directly involved in the control of blood pressure was the study of Morton et al., which showed that neutrophil depletion in normotensive mice, by using an antibody RB6-8C5 to target Gr-1^+^ cells, led to a reduction in systolic blood pressure and to a reduction in endothelial-dependent vasoconstriction [51]. However, this reduction in systolic blood pressure was abolished in the inducible nitric oxide synthase (iNOS) or interferon gamma (IFNγ) KO mice [51], suggesting that, under basal conditions of normotension, neutrophils would contribute to regulating blood pressure, an additional function to the inflammatory defense response. Similarly, the depletion of LysM^+^ cells, a myeloid lineage that involves neutrophils, monocytes, and macrophages, prevented the blood pressure increase induced by AngII infusion [34]. In this study, the adoptive transfer of CD11b^+^ Gr-1^+^ “monocytes” but not of CD11b^+^ Gr-1^+^ “neutrophils” reestablished the AH sensibilization and vascular dysfunction after AngII infusion [34]. The authors discussed that neutrophils could initiate monocyte/macrophage activation, which would promote inflammation and vascular wall dysfunction. However, since Gr-1 is a classical marker for granulocytes, it remains to differentiate the specific effect of neutrophils versus monocytes/macrophages, with a new gating strategy in the flow cytometry analyses.

At the in vitro level, AngII treatment on neutrophils obtained from normotensive patients reduced the abundance and activity of inducible heme oxygenase (HO-1) [47], an antioxidant enzyme that catalyzes the conversion of the heme group, a pro-oxidant group, into biliverdin and then to bilirubin, an antioxidant molecule. Alba et al. found that AngII treatment of “normotensive” neutrophils reduces the nuclear translocation of the transcription factor NF-E2-related factor-2 (Nrf2), a master gene for HO-1 regulation and other antioxidant enzymes [66] (Figure 1). The induction of HO-1 levels was inhibited by AngII in neutrophils isolated from hypertensive patients [47]. Interestingly, the inductions of HO-1 protein and messenger RNA (mRNA) levels were restored in neutrophils of hypertensive patients treated with losartan, a competitive AngII receptor antagonist [47]. Apparently, part of the increase in redox status in AngII-stimulated neutrophils would depend on calcineurin, a phosphatase that is linked to the renal vascular resistance [67].

The hypothesis of ROS imbalance and its relationship with neutrophil activity during AH has also been tested in experimental animals. Neutrophils obtained from spontaneously hypertensive rats (SHR) presented higher levels of iNOS mRNA and a rise in iNOS and MPO enzymatic activity, with a major generation of peroxynitrite and IL-1β [52], demonstrating that neutrophils from SHR rats exhibit an increased formation of ROS and reactive nitrogen species (RNS). Recently, it was shown that iNOS in neutrophils may activate the pathway of leukotriene formation through 5-lipoxagenase, which reinforces the recruitment of immune cells (including neutrophils), favoring their endothelial adhesion [55].

The neutrophil adhesion to the endothelium occurs sequentially in a complex process which, in the first place, depends on the interaction between endothelial selectins (particularly P-selectin) with the P-selectin glycoprotein ligand 1 (PSGL1) [68]. In a second stage, the neutrophil adheres firmly to and crawls along the endothelium through β2 integrins, such as CD11b/CD18, and intercellular adhesion molecule 1 (ICAM-1). In this way, the neutrophil infiltrates the tissue through transcellular and paracellular migrations [68]. The evidence presented suggests an effect of neutrophils on the vascular wall, which can be augmented during AH by the number of circulating cells and by their endothelium adherence for posterior infiltration into the vascular tissue. The endothelium-derived NO decreases the neutrophils’ ability to adhere to the endothelium [69,70] by inhibiting the expression of β2 integrin [71]. Likewise, the exposure of human neutrophils to superoxide anion or hydrogen peroxide generates the induction of CD11b/CD18 adhesion molecules in the neutrophil [72], suggesting that a pro-oxidative microenvironment favors the adhesion and rolling of neutrophils over the endothelium. On the other hand, with regard to the prohypertensive microenvironment, endothelin-1 (ET-1, an endothelial vasoconstrictor) induces the production of IL-8, a neutrophil chemoattractant [73], and increases neutrophil adhesion through ICAM-1 and E-selectin expression in endothelial cells [74,75]. Additionally, Scanzano et al. showed that IL-8 can also increase the mRNA levels of all adrenergic receptors in neutrophils, inducing their chemotaxis and degranulation [76]. However, this effect has not been tested in models of AH, which may be crucial considering the possible role that the autonomous system would have on the AH. Finally, ET-1 may induce CD11b/CD18 in neutrophils in an ET-1 receptor-dependent manner [75].

On the basis of the above evidence, neutrophils could contribute to the control of baseline blood pressure and participate in the AH development by releasing ROS and RNS, favoring the generation of vascular dysfunction.

### 3.2. Neutrophils Can Induce Tissue Inflammation and Fibrosis

AH has been cataloged as a chronic subinflammatory state that, in the long term, induces target organ damage. In 2014, Wu et al. described the participation of S100a8/a9 proteins in the cardiac inflammation driving fibrosis at an early stage of AH [53]. S100a8/a9 proteins are members of the S100 calcium-binding family of proteins primarily expressed in myeloid cells, such as neutrophils, having multiple and complex functions that include NADPH oxidase activation, phagocytosis, migration, and adhesion [77]. By using microarray analysis from the heart, after 24 h of AngII infusion, the authors found an upregulation of S100a8/a9 that gradually decreased after 7 days [53]. Interestingly, circulating neutrophils (CD11b^+^Gr-1^high^) in mice treated for 24 h with AngII showed an induction in S100a8/a9 mRNA abundance [53]. In the same study, cardiac fibroblasts presented a higher abundance of receptor for advanced glycation end products and for Toll-like receptor-4 mRNA, which act as receptors for S100a8/a9. Finally, cardiac fibroblasts presented a proinflammatory state, represented by the induction of different cytokines (IL-1β, IL-6, IL-12p40 subunit, and TNF-α), after treatment with recombinant S100a8/a9 [53]. This suggested a potential role of secreted S100a8/a9 from neutrophils in the modulation of proinflammatory processes driving fibrosis.

On the other hand, anti-S100a9 administration suppressed the infiltration of leukocytes, monocytes, and neutrophils at the cardiac level, in accordance with the prevention of the increase in IL-1β and TNF-α mRNAs (cytokines), as well as CCL2, CCL3, CCL5, and CCL7 mRNAs (chemokines), after 24 h of AngII infusion [53]. In addition, the S100a9 blockade prevented perivascular and interstitial fibrosis, myocardial hypertrophy, and the cardiac induction of profibrotic genes (α-SMA, TGF-β1, collagen-I, and collagen-III) and proinflammatory genes (IL-1β, IL-6, and CCL2) after 7 days of AngII administration [53]. However, anti-S100a9 did not present any effect on AH, in comparison to the isotype control group. The authors concluded that S100a8/S100a9 proteins are key molecules for the onset of cardiac inflammation that leads to fibrosis during AngII administration, but these effects may be independent of blood pressure response. Kerkhoff et al. reported that neutrophils from S100a9 KO animals show less NADPH oxidase activity, suggesting its possible involvement in regulating ROS levels. Apparently, S100a8/a9 allows the binding between arachidonic acid and the NADPH gp91phox subunit, favoring its oxidase activity [78]. On the other hand, it has been demonstrated that the exogenous administration of S100a8/a9 on endothelial cells increases the expression of adhesion molecules, such as ICAM-1, promoting cell permeability [79]. Overall, although there is no direct relationship between S100a8/a9 and AH, all these antecedents suggest that it contributes to pro-oxidant stress, promoting vascular infiltration, which could also favor the adhesion of other immune cells. New experiments that consider the S100a8/a9 modulation in neutrophils are necessary, in order to evaluate their role in the endothelial phenotyping and in AH.

It is well accepted that, during AH, renal and CV injuries are highly dependent on leukocyte recruitment and infiltration. Apparently, DC–neutrophil communication, mediated by different chemokines (CCL3, CCL4, CCL5, and CXCL1) [80] and their receptors, could also participate in endothelial dysfunction and AH development [81]. In this sense, Ahadzadeh et al. tested whether the fractalkine CXCL1 receptor (CX_3_CR_1_), a leukocyte surface receptor involved in chemoattraction, adhesion, and inflammation [54], was involved in renal injury during AH; they found that CX_3_CR_1_ ablation increased glomerular and tubulointerstitial damage and interstitial fibrosis induced by AngII, where some of these effects were dependent on a decrease in renal DCs and on an increase in macrophages and neutrophils (Ly6G^+^ CD11b^+^ cells) in AngII-dependent AH [54]. This increased number of neutrophils was corroborated in interstitial tissue and explained as a secondary response to the glomerular and tubulointerstitial injuries [54]. Since CX_3_CR_1_ is mainly expressed on DCs, these findings suggest an interrelation between DCs and neutrophils during the development of hypertensive-induced renal inflammatory damage.

In the 2000s, a protein called neutrophil gelatinase lipocalin-associated (NGAL) emerged, which has been closely linked to AH in recent years. NGAL, also known as lipocalin 2 (Lcn2), binds siderophores [82], conferring it an important role in the innate immune response against bacterial infections. NGAL is expressed in moderate abundance in the kidney [83] and in diverse cellular types of the CV system [84], with a relatively high expression in hematopoietic organs, neutrophils, and antigen-presenting cells [85,86]. A clinical study performed in patients with essential AH revealed that serum NGAL levels are increased twofold in comparison to healthy normotensive subjects [87]. Furthermore, polymorphisms in the human NGAL promoter have been correlated with high blood pressure levels [88]. In a recent study, it was shown that NGAL is necessary for the development of AH and CV fibrosis in NAS mice [89], while Buonafine et al. showed that NGAL absence, particularly in myeloid cells, is sufficient to prevent the hypertensive phenotype after NAS stimulus [86]. Recently, we demonstrated that NGAL ablation prevents the overexpression of the IL-23p19 subunit, a polarizing/maintaining cytokine for the T_H_17 phenotype, after aldosterone stimulation on DC cultures [32]. However, whether NGAL presents some direct regulation of the secretion of cytokines, chemokines, or ROS from neutrophils during AH remains unknown. This preclinical evidence suggests that NGAL in some immune states may be crucial for the development of AH. However, it is necessary to determine whether NGAL on neutrophils has a direct effect on blood pressure.

In conclusion, under hypertensive conditions, neutrophils infiltrate the arteries, heart, and kidney, promoting the recruitment of other immune cells for tissue inflammation, through secretion of pro-oxidant, pronitrant, and proinflammatory molecules driving fibrosis. However, it remains unclear whether this process continues over time or is crucial only in the early stages of AH.

## 4. Conclusions

Neutrophil function appears to go beyond its phagocytic role in response to pathogen infection. In this review, we presented the clinical and preclinical evidence available regarding the possible role of neutrophils in AH, including different mechanisms that could promote vasoconstriction. In this sense, vascular damage caused by ROS is the most plausible mechanism to date that could explain the relevance of neutrophils in AH. However, since there is an adrenergic regulation of neutrophils, this mechanism should also be considered, in addition to catecholamine and acetylcholine secretion from neutrophils, which may modulate the vascular tone during AH.

Neutrophils could also apparently be involved in the inflammatory processes of other CV diseases such as atherosclerosis, myocardial infarction, and stroke [90,91]. Granulocyte recruitment during ischemia/reperfusion in the heart of WT mice was shown, which was later associated with the modulation of chemokines that drive infiltration and adhesion [92]. In addition, neutrophil extracellular traps (NETs) have recently been linked to different CV diseases, such as arteriosclerosis [93], ischemic stroke [94], thrombotic stroke [95], acute coronary syndrome [96], and others. However, there is no direct evidence of the participation of NETs in AH. Since AH is a risk factor for the aforementioned CV events, and since there is an increase in neutrophil count in AH, the possible relationship between NETs and AH deserves to be elucidated.

In relation to the role of neutrophils in hypertensive kidney disease, there is less evidence supporting a direct link. However, it is well accepted that vascular damage during AH drives renal dysregulation, where neutrophils are also accumulated in both glomerular and tubulointerstitial compartments as part of the proinflammatory effects, with an important role in fibrosis development [97].

Therefore, neutrophils may infiltrate target organs, secreting proinflammatory molecules and vasoactive neurotransmitters, triggering the onset of the characteristic subinflammatory state in AH or favoring organ dysfunction due to the recruitment of other immune cells in the long term (Figure 2). Importantly, neutrophils modulate the immune response of T lymphocytes, macrophages, and DCs, which have been proposed as crucial cell types for the onset of experimental AH. However, the possible causal role of neutrophils and AH has not yet been elucidated, and future studies are necessary to test this hypothesis.

## Figures and Tables

**Figure 1 ijms-21-08536-f001:**
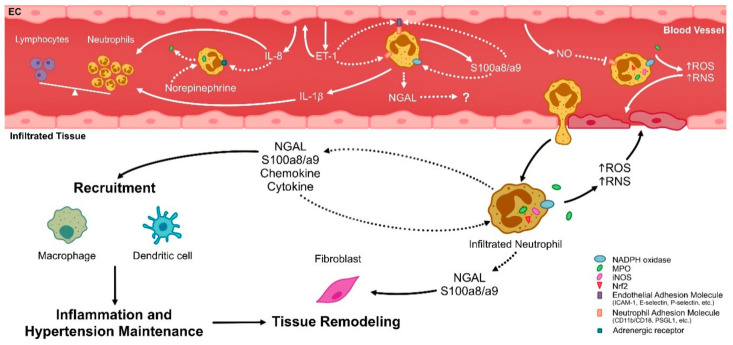
Possible mechanisms of neutrophils in the pathophysiology of AH. In arterial hypertension (AH) there is an increase in circulating neutrophils, represented as total neutrophil count or neutrophil/lymphocyte ratio (NLR), which can be influenced by interleukin (IL)-1β secreted by activated neutrophils and IL-8, which in turn is secreted from endothelial cells (ECs) and induced by endothelin-1 (ET-1). In addition, IL-8 can induce adrenergic receptors in neutrophils, promoting their activation. ET-1 would increase the expression of adhesion molecules favoring the EC–neutrophil interaction and adhesion process. In the same way, S100a8/a9 secreted by neutrophils may induce adhesion molecules in ECs. Adhesion and rolling process are promoted by reactive oxygen species (ROS) and inhibited by nitric oxide (NO) derived from ECs. Circulating neutrophils, as well as neutrophils that infiltrate tissue, generate ROS and reactive nitrogen species (RNS) through enzymes such as Nicotinamide adenine dinucleotide phosphate (NADPH) oxidase, myeloperoxidase (MPO) which can be released by adrenergic stimulation, and inducible nitric oxide synthase (iNOS) during the AH. In addition, NF-E2-related factor-2 (Nrf-2) is retained in the cytosolic compartment of neutrophils, avoiding the induction of antioxidant enzymes. Together, this oxidative and nitrative stress may cause endothelial damage, inducing vascular dysfunction and AH. On the other hand, neutrophils promote tissue damage and immune cell recruitment, such as macrophages and dendritic cells, through the secretion of cytokines, chemokines, neutrophil gelatinase-associated lipocalin (NGAL), or S100a8/a9. In turn, these molecules may induce an autocrine effect on blood neutrophils, promoting their activation, as well as on infiltrated tissues, promoting remodeling processes perpetuating the AH. Continuous and intermittent lines represent demonstrated and hypothesized mechanisms in AH, respectively.

**Figure 2 ijms-21-08536-f002:**
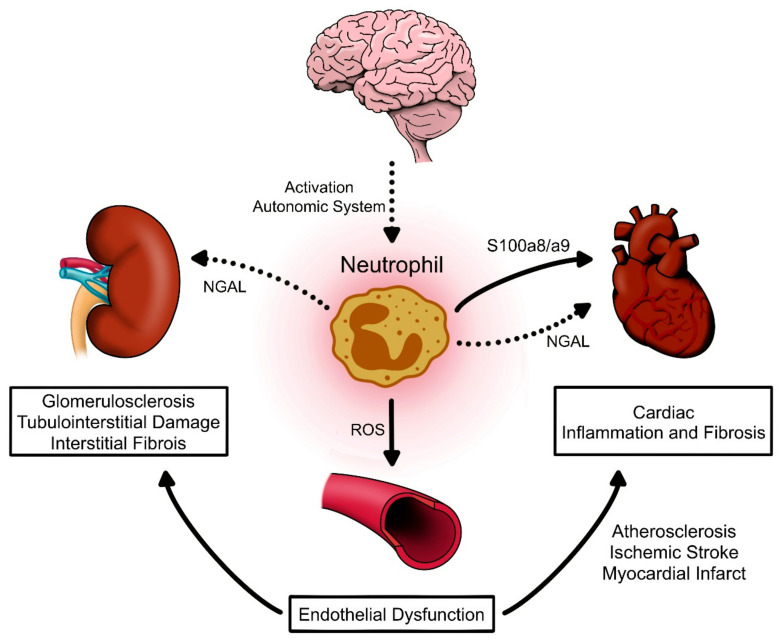
Neutrophil participation in cardiovascular and renal damage related to the hypertensive phenotype. During arterial hypertension (AH), where the autonomic system can activate neutrophils, neutrophils may induce vasculature damage, thereby secreting reactive oxidative species (ROS) promoting endothelial dysfunction. Once in target organs, neutrophils can contribute to inflammation, as well as cardiac fibrosis, atherosclerosis, ischemic stroke, and myocardial infarction, driving heart failure in part through NGAL and S100a8/a9. Neutrophils can also infiltrate the kidneys in AH stimulating glomerulosclerosis, tubulointerstitial damage, and interstitial fibrosis, which are features of hypertensive kidney damage that perpetuate kidney dysfunction and that may be favored by NGAL–neutrophil secretion.

**Table 1 ijms-21-08536-t001:** Studies in patients and experimental animals, involving neutrophils in blood pressure.

Experimental Model	Main Findings Related to Neutrophils and Blood Pressure	Reference
Studies in Patients or Human Samples		
**Blood neutrophils from AH patients** *N* = 37; case-control study	Superoxide production was increased in neutrophils from AH patients	[44]
**Blood neutrophils from women with preeclampsia** *N* = 34; case-control study	Superoxide production increased in neutrophils from women with preeclampsia	[45] [46]
**Blood neutrophils from AH patients** *N* = 9,383; cohort study *N* = 28,850; cohort study *N* = not indicate	Neutrophils were associated with incidence of AH and correlated with more risk of AH	[38] [39]
	Reduction of HO-1 and Nrf2 in neutrophils from AH	[47]
**Blood neutrophils from AH patients** *N* = 72; double-blind randomized prospective study *N* = 46; randomized study	Nebivolol and Valsartan decreased NLR ratio in AH patients	[42] [43]
*N* = 166; cross-sectional study *N* = 409; single-center retrospective study	NLR and neutrophil count were increased in AH patients with non-dipper pattern	[40] [48]
*N* = 150; observational study *N* = 33; cross-sectional study	NLR and neutrophil count were increased in RHT patients and in patients with ’normal-high’ AH grade	[41] [49]
**Blood neutrophils from elderly patients with AH** *N* = 341 single-center observational study	NLR was linked to a high probability of mortality in elderly patients with AH	[50]
**Studies in Experimental Animals**		
Neutrophil depletion in normotensive mice	BP reduction in normotensive WT mice; iNOS or IFNγ ablation reversed this effect	[51]
SHR	iNOS, MPO activity and IL-1β increased in circulating neutrophils	[52]
AngII infused mice	Circulating and aortic wall neutrophils increased in AH mice	
	Depletion of LysM^+^ cells prevented AH and increased circulating neutrophils	
	Adoptive transfer of neutrophils did not reestablish AH	[34]
AngII infused mice	Early induction of S100a8/a9 in circulating neutrophils	[53]
	The anti-S100a9 suppressed heart infiltration of neutrophils, with no effect on BP	
Nephrectomy-AngII-salt mice	CX3CR1 ablation induced high renal damage with increased neutrophils infiltration on kidney	[54]
L-NAME hypertensive mice	Increased leukotrienes in neutrophils supernatants isolated from L-NAME mice	[55]

AngII, angiotensin II; AH, arterial hypertension; BP, blood pressure; CX3CR1, CXCL1 receptor; HO-1, heme oxygenase 1; iNOS, inducible nitric oxide synthase; IFNγ, interferon gamma; IL, interleukin; LysM, lysozyme M; MPO, myeloperoxidase; Nrf-2, NF-E2-related factor 2; NLR, neutrophil/lymphocyte ratio; L-NAME, *N*-nitro-l-arginine methyl ester; RHT, resistant hypertension treatment; SHR, spontaneously hypertensive rats; WT, wild type.

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
