# Peer review of "The Role of Neutrophils in Hypertension"

_ijms, 2020, doi:10.3390/ijms21228536_

Round 1

Reviewer 1 Report

This is an interesting review that highlights the emerging link between neutrophils and the development of hypertension as well as cardiorenal fibrosis. This review is well conceived and structured, and provides current knowledge in this area (including authors' own work). I recommend this paper for publication with only one minor comment to address. In order to comprehensively address this topic, I suggest to cite and discuss the recent paper by Siedlinski et al. which was published on Circulation in 2020. In that paper, Siedlinski et al have demonstrated that lymphocytes but not neutrophils are causally related to human blood pressure using mendelian randomization analyses.

Trivial

Page4, line 167, “that” is duplicated

Author Response

We really appreciate the commentaries done by the reviewer.

- We added a new paragraph with the discussion of the work done by Siedlinski et al., according the recommendation (line 142 in the new version). We hope that this new information may permit a better understanding of this section.

- We have deleted the duplicated word in the new version of manuscript.

Reviewer 2 Report

This review of the role of neutrophils in hypertension will be an important addition to the literature on this topic. It nicely summarized the work done in this area with a number of useful references. I felt that the review made such a good case for the role of neutrophils that they sucj consider whether the word "potential" should be omitted from the title.

Author Response

We appreciate the comment done by the reviewer.

We corrected the main title according the recommendation.